# A new insight into RecA filament regulation by RecX from the analysis of conformation-specific interactions

Aleksandr Alekseev[1]*[†], Georgii Pobegalov[1]*[†], Natalia Morozova[1], Alexey Vedyaykin[1], Galina Cherevatenko[1], Alexander Yakimov[1], Dmitry Baitin[2], Mikhail Khodorkovskii[1]

[1]Peter the Great St. Petersburg Polytechnic University, St. Petersburg, Russian Federation; [2]Petersburg Nuclear Physics Institute Named by B.P. Konstantinov of National Research Center «Kurchatov Institute», Gatchina, Russian Federation

**Abstract** RecA protein mediates homologous recombination repair in bacteria through assembly of long helical filaments on ssDNA in an ATP-dependent manner. RecX, an important negative regulator of RecA, is known to inhibit RecA activity by stimulating the disassembly of RecA nucleoprotein filaments. Here we use a single-molecule approach to address the regulation of (*Escherichia coli*) RecA-ssDNA filaments by RecX (*E. coli*) within the framework of distinct conformational states of RecA-ssDNA filament. Our findings revealed that RecX effectively binds the inactive conformation of RecA-ssDNA filaments and slows down the transition to the active state. Results of this work provide new mechanistic insights into the RecX-RecA interactions and highlight the importance of conformational transitions of RecA filaments as an additional level of regulation of its biological activity.

## Editor's evaluation

This paper is of interest to readers in the fields of DNA repair, DNA-protein interactions, and those employing single-molecule techniques. Using single-molecule methods, the authors visualized how RecX, a negative regulator of homologous recombination in bacteria, interferes with the active species in recombination, the RecA nucleoprotein filament. They showed that RecX binds to the RecA filament in its post-ATP hydrolysis state, promotes RecA dissociation from ssDNA, and causes a reversible conformational change of the filament. The latter mode of RecX action is novel and of particular interest. The authors present an interesting model of the RecX-RecA-ATP-ssDNA system.

*For correspondence:
a.alekseev@nanobio.spbstu.ru (AA);
george.pobegalov@nanobio.spbstu.ru (GP)

[†]These authors contributed equally to this work

Competing interest: The authors declare that no competing interests exist.

## Introduction

In bacteria, the RecA protein is a central player in DNA repair, genetic recombination, and SOS response activation (*Cox, 1999*; *Lusetti and Cox, 2002*; *Cox, 1991*; *Clark and Sandler, 1994*; *d'Ari, 1985*). RecA acts through assembly of long helical filaments on ssDNA in an ATP-dependent manner (*van Loenhout et al., 2009*; *VanLoock et al., 2003b*; *Takahashi and Nordén, 1994*; *Galletto et al., 2006*; *Hegner et al., 1999*). RecA-ssDNA filaments can further pair with homologous duplex DNA, catalyzing DNA strand exchange reaction, a key step during homologous recombination which ensures rescue of stalled replication forks, faithful DNA repair, and genetic recombination (*Cox, 2007a*; *Heller and Marians, 2006*). Apart from that, RecA-ssDNA filaments stimulate proteolytic cleavage of the LexA repressor, allowing activation of over 40 SOS response genes involved in DNA repair and cell cycle regulation (*Little, 1993*; *Baralla, 2008*). Importantly, RecA-dependent SOS response activation is one of many pathways for development of antibiotic resistance by bacteria through enhanced

mutagenesis in the presence of continuous DNA damage. RecA nucleoprotein filaments are very dynamic and may experience large-scale conformational changes induced by ATP binding and hydrolysis (*van Loenhout et al., 2009*; *Kim et al., 2014*; *Kim et al., 2017*; *Nishinaka et al., 2007*; *Alekseev et al., 2020b*). Besides intrinsic ATPase activity of RecA, assembly and stability of RecA-ssDNA filaments are also dynamically regulated by a network of various protein mediators (*Bell et al., 2012*; *Gasior et al., 2001*; *Morimatsu and Kowalczykowski, 2003*; *Cox, 2007b*; *Baitin et al., 2008*).

RecX is an important negative regulator, which has been reported to suppress ATPase, DNA pairing, and strand exchange activities of RecA (*Ragone et al., 2008*; *VanLoock et al., 2003a*; *Le et al., 2017*; *Cárdenas et al., 2012*; *Gruenig et al., 2010*). RecX also inhibits RecA co-protease activity (*Stohl et al., 2003*). Genes encoding RecX were found in genomes of a wide diversity of bacteria and some plants (*Lin et al., 2007*). In *Escherichia coli*, recX is a SOS-regulated gene located downstream of recA, which encodes a small 19.4 kDa protein. recX and recA are co-transcribed; however, expression of recX is downregulated at both transcriptional and translational levels resulting in about a 500-fold lower protein level compared to RecA (*Pagès et al., 2003*).

In vivo studies showed that either loss-of-function RecX mutation or overexpression of RecX decreases bacterial resistance to UV irradiation (*Stohl et al., 2003*), while overexpression of RecA had a toxic effect on cell viability when RecX was mutated (*Vierling et al., 2000*; *Papavinasasundaram et al., 1998*; *Sano, 1993*; *Stohl and Seifert, 2001*). For some species, such as *Deinococcus radiodurans*, RecX shows dual-negative regulation of RecA function - it not only directly inhibits RecA activity at the protein level but also inhibits RecA induction at the transcriptional level (*Sheng et al., 2005*). Interestingly, RecX is also an important mediator of natural transformation in *Bacillus subtilis*, where it colocalizes with RecA threads, while the lack of RecX decreases chromosomal transformation approximately by 200-fold (*Cárdenas et al., 2012*).

A capping model has been proposed to elegantly explain the inhibitory mechanism of RecX at substoichiometric concentrations relative to RecA. According to this model, RecX binds to the growing 3'-end of the RecA filament and blocks the filament extension, which results in net filament depolymerization (*Drees et al., 2004a*). On the other hand, observation of faster RecA depolymerization at higher RecX concentrations together with structural evidence of RecX binding along the RecA filament groove (*Ragone et al., 2008*; *VanLoock et al., 2003a*; *Shvetsov et al., 2014*) indicates that alternative internal nicking mechanism may exist, in which RecX locally destabilizes the RecA filament and increases the number of the disassembling ends (*Gruenig et al., 2010*; *Venkatesh et al., 2002*). In support of this model, a recent study of *Mycobacterium smegmatis* RecX (MsRecX) showed that incubation of MsRecX with RecA filaments resulted in RecA dissociation from within the filament. It has also been shown that mechanical forces, which are an important factor in regulating the stability of the RecA filament (*Le et al., 2017*; *Fu et al., 2013*), can counteract the inhibitory effect of RecX at as little as 7 pN, prevent disassembly of the RecA filament, and even stimulate the repolymerization of RecA on DNA in the presence of RecX.

Another completely different model of the RecX inhibitory action was proposed based on the transmission electron microscopy data (*VanLoock et al., 2003a*). This model is not associated with a decrease in the number of RecA monomers on DNA as a result of filament depolymerization, but alternatively suggests that RecX binds to the interface between RecA monomers and blocks the transition between inactive and active states, thereby preventing ATP hydrolysis.

Single-molecule studies proved to be a powerful tool for addressing the inhibitory mechanism of RecX. Magnetic tweezers assays have been used to directly observe disassembly of RecA filaments induced by RecX proteins from *Mycobacterium tuberculosis* (*Le et al., 2014*) and *B. subtilis* (*Le et al., 2017*). Measurements of the dynamic characteristics of RecA-ssDNA filaments in the active state showed that RecX promotes net depolymerization of preformed filaments at low tensile forces (about 3 pN) in ATP hydrolysis and RecX concentration-dependent manner, which is generally consistent with results of ensemble experiments. At the same time, this approach made it possible to discover that depolymerization of the RecA filament took place in a non-monotonic stepwise manner with pauses of various lengths (10–100 s). Moreover, RecA depolymerization could proceed with an initial lag phase followed by a strikingly rapid net depolymerization phase (*Le et al., 2017*) indicating a complicated nature of interaction between RecX and RecA-DNA complex.

Recently, we characterized three mechanically distinct conformational states that occur within the *E. coli* RecA-ssDNA filaments in the course of ATP hydrolysis (*Alekseev et al., 2020b*). In the present

work, building on the developed single-molecule assay we aimed to study interactions of *E. coli* RecX with different forms of RecA nucleoprotein filaments to further deepen understanding of RecX inhibitory mechanism.

## Results

### RecX stimulates biphasic shortening of RecA-ssDNA filaments

In this work, we addressed the dynamics of interaction between *E. coli* RecX and RecA nucleoprotein filaments (*Figure 1A*) utilizing a single-molecule approach which combines dual-trap optical tweezers and a microfluidic laminar flow cell for manipulation of individual DNA molecules (*Figure 1B*) as reported elsewhere (*Alekseev et al., 2020b*; *Candelli et al., 2014*; *Gutierrez-Escribano et al., 2019*; *Forget and Kowalczykowski, 2012*; *Brouwer et al., 2016*; *Belan et al., 2021*; *Alekseev et al., 2020a*; *Yakimov et al., 2017*). The microfluidic flow cell (Lumicks B.V.) consisted of five laminar flow channels which facilitated optical trapping of the beads, DNA tethering, and RecA nucleoprotein filaments assembly (see Materials and methods). Additionally, changing the composition and order of the microfluidic channels provided flexibility in the experimental design to study how RecX interacts with various forms of RecA-DNA filaments (*Figure 1—figure supplement 1*).

Previously, single-molecule magnetic tweezers experiments revealed that *M. tuberculosis* RecX and *B. subtilis* RecX stimulate gradual disassembly of the corresponding active RecA-ssDNA filaments resulting in the net decrease of the DNA tether length (*Le et al., 2017*; *Le et al., 2014*). First, we aimed to investigate whether *E. coli* RecX (further referred as RecX) exhibits the same behavior. To address the dynamics of RecA filaments preassembled on 11,071 nt long ssDNA (further referred as RecA-ssDNA filaments), the RecA-ssDNA tether was stretched with a constant force of 3 pN while its end-to-end distance was simultaneously recorded. The force was chosen as low enough to not introduce structural deformations in the filament as reported previously (*Alekseev et al., 2020b*).

Introduction of the preassembled RecA-ssDNA filament into the channel containing RecX (0.2–1 µM) in the presence of both RecA (1 µM) and ATP (1 mM) resulted in the initial step-like shortening of the tether followed by a slow decrease in the end-to-end distance (*Figure 1C*). After 200 s of incubation with RecX the length reduction reached steady state and proceeded with an almost linear profile. Average rate of the filament length reduction was dependent on the concentration of RecX (*Figure 1D*). Higher RecX concentrations stimulated faster shortening in line with previously reported observations for *M. tuberculosis* and *B. subtilis* RecX (*Le et al., 2017*; *Le et al., 2014*). Interestingly, the initial step-like shortening of the RecA-ssDNA filament also scaled with RecX concentration (*Figure 1E*). Corresponding dependence is presented in *Figure 1E* and is well fitted by a Hill equation with a Hill coefficient of 2.0±0.3. These observations indicate that *E. coli* RecX stimulates shortening of the RecA-ssDNA filaments which proceeds in two phases.

### Shortening of RecA-ssDNA filaments induced by RecX is driven by reversible conformational change in the filament and slow filament disassembly

At a stretching force of 3 pN, the length of the RecA-ssDNA filament is significantly greater than the length of bare ssDNA (*van Loenhout et al., 2009*; *Alekseev et al., 2020b*; *Figure 1—figure supplement 2*). Therefore, shortening of the RecA-ssDNA tether can be caused by RecX displacing RecA from ssDNA. Alternatively, since both RecX and RecA were present at comparable concentrations, RecX could interact with free RecA in solution preventing its interaction with DNA.

Additionally, RecA-ssDNA filaments may adopt different conformations depending on the nucleotide cofactor bound in the RecA monomer-monomer interface. ATP-bound or 'active' form is characterized by a longer pitch which results in a stretched conformation of the filament (*Yu and Egelman, 1992*; *DiCapua et al., 1990*; *Lebedev et al., 2003*; *Ellouze et al., 1995*; *Ellouze et al., 1999*; *Stasiak et al., 1988*; *Chen et al., 2008*), while 'inactive' ADP-bound and *apo* forms are characterized by a smaller pitch and a more compact structure (*Yu and Egelman, 1992*; *DiCapua et al., 1990*; *Lebedev et al., 2003*; *Ellouze et al., 1995*; *Ellouze et al., 1999*; *Ruigrok et al., 1993*; *Chang et al., 1988*). Hence, shortening of the RecA-ssDNA tether induced by RecX can be caused not only by RecA dissociation but also by a conformational transition between active and inactive states.

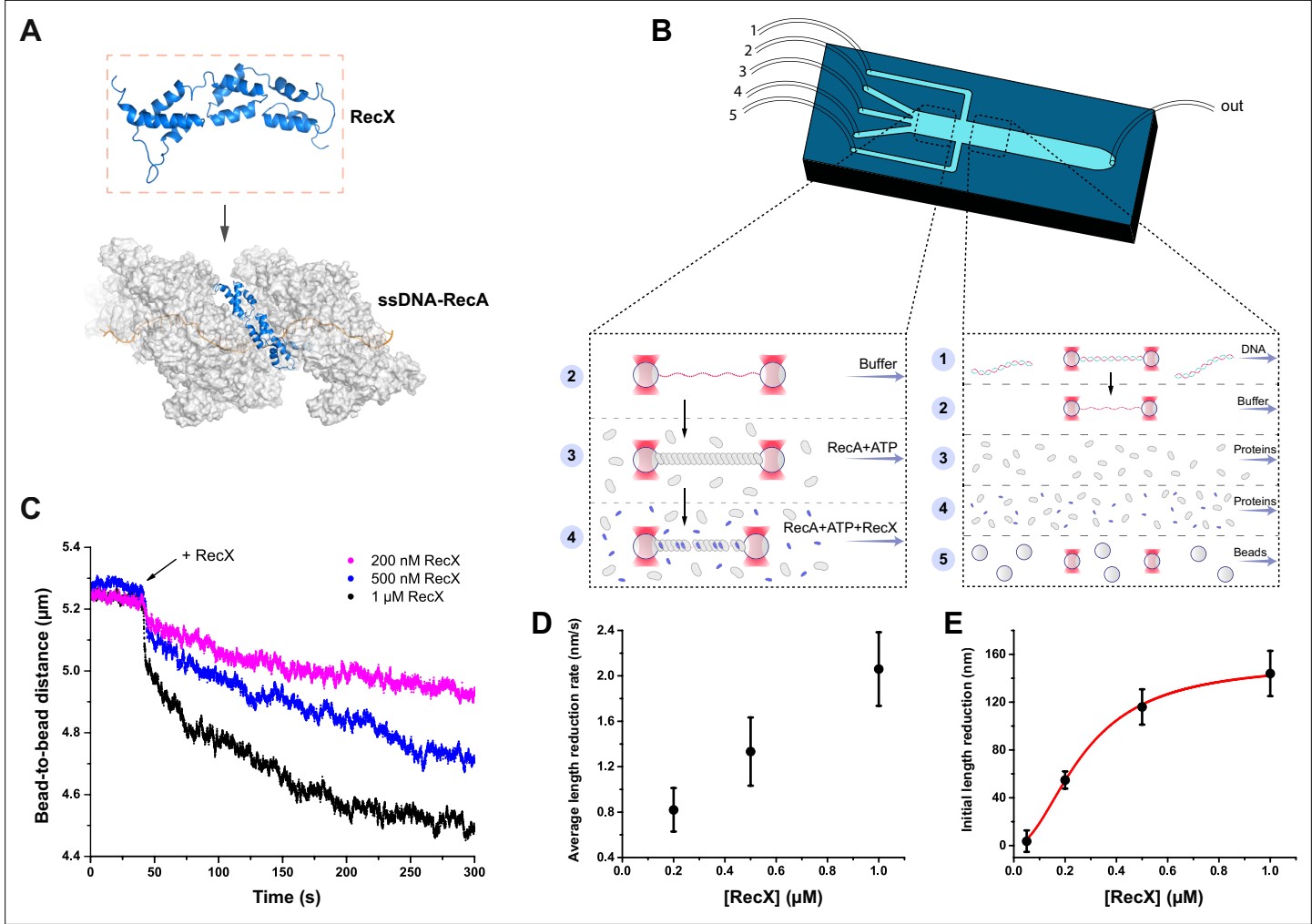

**Figure 1.** The study of the RecX effect on the RecA-ssDNA filaments. (**A**) A schematic of RecX binding along the groove of the active RecA-ssDNA. Atomic structure model for RecA::RecX::ssDNA is adopted from *Shvetsov et al., 2014*. (**B**) A schematic of a five-channel microfluidic flow cell (Lumicks). Dash line highlights two working regions. The three-channel region was used to study the effect of RecX on the RecA-ssDNA filament. In the five-channel region, the beads trapping, DNA tether formation, and generation of ssDNA by force-induced melting were performed. (**C**) The change in the length of RecA-ssDNA filament upon transition from the channel containing 1 µM RecA and 1 mM ATP to the channel containing 1 µM RecA, 1 mM ATP, and various concentrations of RecX. During incubation, a constant tension of 3 pN was applied to the tether. (**D**) The impact of RecX concentration on the average rate of reduction in the RecA-ssDNA filament length over 250 s after initial steep decrease. (**E**) The dependence of the RecX induced initial sharp decrease in RecA-ssDNA filament on the RecX concentration. Solid curve - fit of experimental data with Hill equation with a Hill coefficient of 2.0±0.3. Each data point in (**D**) and (**E**) is a mean value of at least three measurements, bars represent SD.

The online version of this article includes the following source data and figure supplement(s) for figure 1:

**Source data 1.** Source data for traces of RecA-ssDNA filaments, average length reduction rate, and initial length reduction values.

**Figure supplement 1.** Single-molecule assay.

**Figure supplement 1—source data 1.** Source data for *Figure 1—figure supplement 1E*.

**Figure supplement 2.** The comparison of force-extension behavior of bare ssDNA (blue) and the ATP-bound RecA-ssDNA filament (black).

To further elucidate the details of the RecX interaction with RecA nucleoprotein filaments, we performed experiments without the presence of free RecA during the incubation with RecX. For this, we first preassembled the RecA-ssDNA filament in the channel containing RecA and ATP and transferred the tether into a channel containing only the buffer supplemented with ATP (*Figure 2A*). Upon transfer, RecA-ssDNA tether did not show any significant change in its length and was relatively stable. This confirmed that RecA is tightly bound to ssDNA, and no turnover between DNA-bound and free RecA is necessary to keep RecA-ssDNA filaments stable.

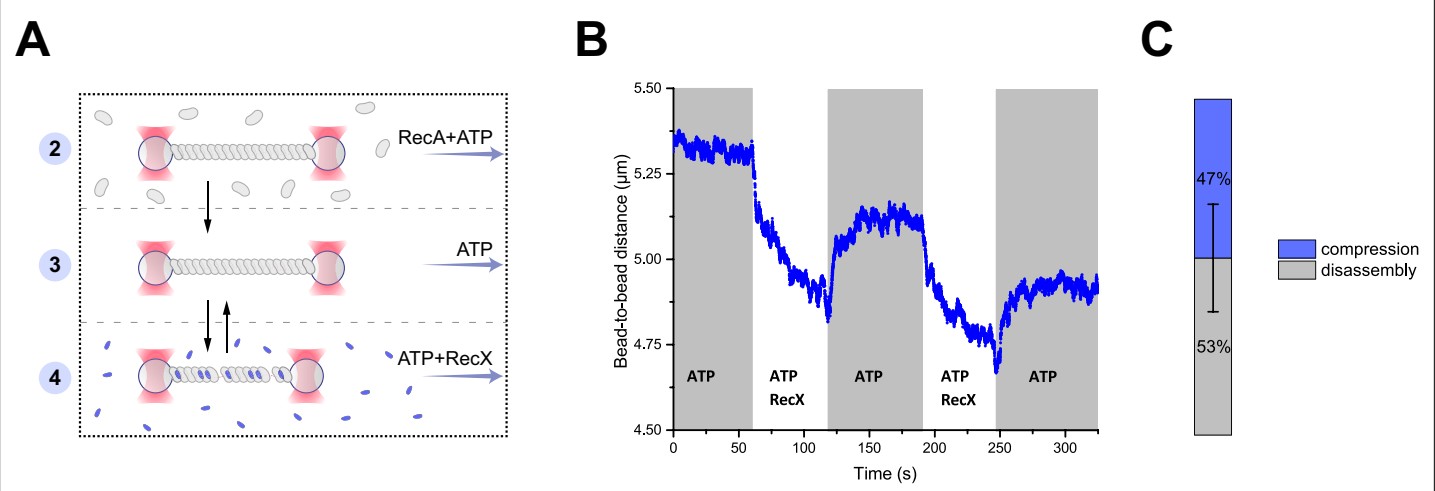

**Figure 2.** RecX-induced reversible changes in the RecA-ssDNA filament structure. (**A**) A schematic of the experiment revealing that RecX is able to induce reversible structural changes in the RecA-ssDNA filaments. (**B**) RecX induces reversible changes in the RecA-ssDNA filament structure in the presence of ATP. (**C**) A comparison of the reversible (compression) and the irreversible (disassembly) reduction in RecA-ssDNA filament length. Stacked histogram represents multiple measurements for six different molecules. Bars represent SD.

The online version of this article includes the following source data for figure 2:

**Source data 1.** Source data for RecX-induced reversible changes in the length of RecA-ssDNA filament.

Next, the RecA-ssDNA filament was transferred into a channel containing ATP (1 mM) and RecX (1 µM). This transition resulted in the reduction of the tether length (*Figure 2B*) in a manner similar to the experiment described in the previous section, indicating that shortening of the RecA-ssDNA filament is induced by direct interaction of RecX with the DNA-bound RecA since this process does not depend on the presence of free RecA. Surprisingly, when the RecA-ssDNA filament was moved back to the channel containing ATP without RecX a gradual elongation of the tether length was observed (*Figure 2B*). Since no free RecA was present, the observed elongation is solely driven by structural rearrangement within the RecA-ssDNA filament. This indicates that binding of RecX to the RecA-ssDNA tether introduces a conformational change within the RecA nucleoprotein filament that leads to a shortening of its overall length, which can be reversed when RecX is eliminated.

Cycles of shortening and elongation of the tether could be observed multiple times on the same DNA molecule by simply transferring the RecA-ssDNA filament between the RecX and the buffer channels (*Figure 2B*). However, after each incubation with RecX subsequent length restoration was not complete with the RecA-ssDNA filaments consistently showing decreased length. This indicates that at each cycle a part of RecA irreversibly dissociated from DNA (*Figure 2C*). These observations suggest that RecX-induced shortening of the RecA-ssDNA tether is a combined result of two processes: a fast conformational transition into a more compact form and slow disassembly of the RecA nucleoprotein filament.

## RecX interacts with inactive RecA-ssDNA filaments and inhibits transition into the active state

At saturated concentrations of ATP (1 mM used in the current work), the RecA-ssDNA filament adopts an overall 'active' or stretched conformation. However, continuous ATP hydrolysis occurs throughout the filament and generates a dynamic and heterogeneous structure in which *apo* and ADP conformations may transiently occur as local patches (*Alekseev et al., 2020b*; *Alekseev et al., 2020a*; *Boyer et al., 2019*). One possible explanation of the reversible shortening of the RecA-ssDNA tether is that RecX interacts with the compressed conformations locally occurring within the filament and increases their lifetime. To test this hypothesis, we addressed the interaction of RecX with the *apo* form of RecA-ssDNA filaments.

We first assembled an active filament in the presence of 1 µM RecA and 1 mM ATP and then transferred it into a channel containing a buffer without ATP (*Figure 3A*). ATP hydrolysis stimulated

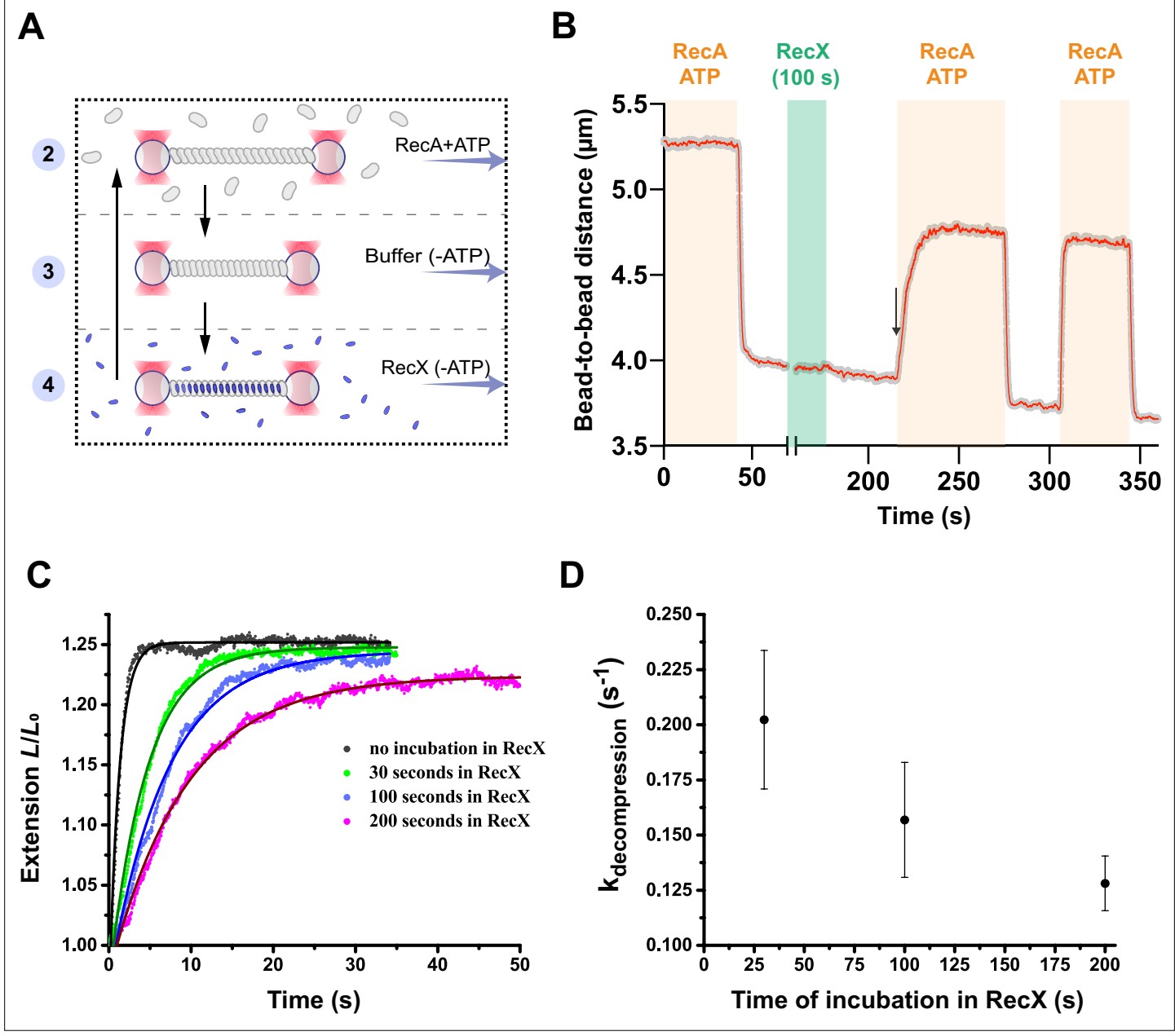

**Figure 3.** RecX affects the conformational transition of RecA-ssDNA filament from the inactive state to the active state. (**A**) A schematic of the experiment revealing that RecX binds inactive RecA-ssDNA filaments. (**B**) The change of the RecA-ssDNA filament length upon conformational transitions between *apo* and ATP-bound states. Incubation of *apo* RecA-ssDNA filament with 500 nM RecX (green area) leads to a slowdown of the subsequent decompression of the RecA-ssDNA filament (black arrow points the beginning of the slowed down decompression). A constant tension of 3 pN was applied to the tether during incubation and transitions. (**C**) Relative extension of the RecA-ssDNA filament in the course of decompression after incubation of inactive RecA-ssDNA filament with 500 nM RecX for 30, 100, and 200 s. (**D**) Corresponding rate constants of the decompression obtained by exponential fitting (solid line in (**C**)) of the elongation profiles. Each point is a mean of at least six measurements. Bars represent SD.

The online version of this article includes the following source data and figure supplement(s) for figure 3:

**Source data 1.** Source data for slowdown decompression of *apo* RecA-ssDNA filament caused by incubation with RecX.

**Figure supplement 1.** The effect of the slowed down decompression retains when RecA-ssDNA filament is incubated in the RecX-free buffer after short incubation with RecX.

**Figure supplement 1—source data 1.** Source data for *Figure 3—figure supplement 1*.

**Figure supplement 2.** The effect of the slowed down decompression is independent of incubation time of the RecA-ssDNA filament in the *apo* channel in the absence of RecX.

*Figure 3 continued on next page*

*Figure 3 continued*

**Figure supplement 2—source data 1.** Source data for *Figure 3—figure supplement 2*.

**Figure supplement 3.** Incubation of ADP-bound form of the RecA-ssDNA filament with RecX results in the slowdown of the following decompression.

**Figure supplement 3—source data 1.** Source data for *Figure 3—figure supplement 3*.

fast accumulation of the compact *apo* conformation accompanied by a characteristic reduction of the tether length (*Figure 3B*). Next, the filament was transferred into the channel containing 500 nM RecX and incubated for 100 s. During incubation no change in the filament length was registered. After that, the filament was transferred back into the buffer channel and subsequently into the RecA-ATP channel.

In the RecA-ATP channel, we observed a slow elongation of the tether suggesting gradual accumulation of the stretched ATP conformation within the RecA-ssDNA filament due to a conformational transition from the *apo* into the ATP-bound state. However, decompression of the filament after incubation with RecX was remarkably slower compared to the case when incubation with RecX was omitted (*Figure 3C*). Interestingly, when the same molecule was subjected to another cycle of *apo*-ATP transition without RecX incubation, decompression proceeded rapidly with the rate similar to the control case without RecX incubation. This experiment confirms that RecX interacts with the *apo* form of the RecA-ssDNA and delays its conformational transition into the ATP-bound form. However, decompression of the RecA-ssDNA filament irreversibly disrupts a specific linkage between RecX and the *apo* state of the filament.

To quantitatively assess the observed effect, we varied the time of incubation with RecX and calculated the rate constants of the corresponding decompression events using exponential fitting (*Figure 3C and D*). A profile of the filament elongation was fitted with the following expression: $L=L_0+\Delta L \cdot exp^{(-kt)}$, where $L_0$ is the length of the *apo* state of the RecA-ssDNA filament, $\Delta L$ is the change of RecA filament length upon conformational transition from the *apo* to the ATP-bound state, and $k$ is the rate constant of the conformational transition. Increasing the time of incubation with RecX from 30 to 200 s resulted in slowing down of the decompression characterized by decrease in $k_{decompression}$ from 0.2 to 0.13 s$^{-1}$. For comparison, the rate constant of the RecA-ssDNA filament decompression without incubation with RecX was 1.3±0.3 s$^{-1}$ (N=5). This value is limited by the temporal resolution of the current measurement, e.g., the time required to transfer the tether between the channels while operating optical tweezers in a force clamp mode. In the previous work, using the same DNA construct and a fast movement between microfluidic channels the rate constant of the *apo*-ATP transition was estimated to be greater than 10 s$^{-1}$ (*Alekseev et al., 2020b*).

Additional control experiments verified that RecX remains bound to the *apo* form of the RecA-ssDNA filament in the RecX-free buffer (*Figure 3—figure supplement 1*). The observed slowdown in the decompression rate was also independent of how long the RecA-ssDNA filament remained in the inactive state in the absence of RecX (*Figure 3—figure supplement 2*). Our results indicate that RecX tightly binds the *apo* state of RecA-ssDNA filament and stabilizes it, retarding the *apo*-ATP transition. The conformational transition of the RecA-ssDNA filament from the *apo* to the ATP state results in the gradual loss of specific interaction of RecX and the *apo* conformation.

We also examined the interaction of RecX with ADP state of the RecA-ssDNA filament. Recently, it was shown that ADP and *apo* conformations represent two distinct inactive states of RecA-ssDNA filament (*Alekseev et al., 2020b*). Incubation of ADP-bound form of the RecA-ssDNA filament with RecX also resulted in the slowdown of the following decompression (*Figure 3—figure supplement 3*) similar to the RecX interaction with *apo* RecA-ssDNA filament. Interestingly, ADP-bound RecA-ssDNA filaments exhibited greater stability when supplemented with RecX.

## ATP promotes RecX dissociation from RecA-ssDNA filaments

To understand whether RecX remains bound to the RecA-ssDNA filament after ATP-induced decompression we generated a fluorescent version of *E. coli* RecX. For this, the N-terminus of RecX was fused to a green fluorescent protein mNeonGreen (*Clavel et al., 2016*; *Shaner et al., 2013*) via a short peptide linker. mNeonGreen-RecX (RecX$_{mNG}$) retained the inhibitory effect on RecA at the level comparable to the wild-type RecX. This was tested by ability of RecX$_{mNG}$ to inhibit ATPase activity of

RecA in bulk (*Figure 4A*) as well as to slow down the *apo*-ATP transition rate in the single-molecule experiments (*Figure 4B*).

To evaluate binding of RecX$_{mNG}$ to the *apo* form of RecA-ssDNA filaments, we first preassembled the active RecA-ssDNA filament and transferred it into the channel with ATP-free buffer to obtain the *apo* conformation. After that, the tether was incubated in the channel containing 1 µM of RecX$_{mNG}$ for 30 s and then transferred back to the buffer channel (lacking both RecX$_{mNG}$ and ATP) to be imaged using wide-field fluorescence microscopy. Visualization revealed a strong fluorescent signal along the DNA tether confirming that RecX$_{mNG}$ tightly binds the compressed RecA-ssDNA filament (*Figure 4C*). For fluorescence experiments, a longer DNA substrate of ~24,000 nt was used to obtain a larger separation between the beads.

Next, we transferred the RecX$_{mNG}$-bound filament to the channel containing RecA and ATP to induce a transition into the active state and imaged the tether again after decompression was completed. Fluorescent visualization demonstrated significant loss of RecX$_{mNG}$ from the RecA-ssDNA filament (*Figure 4C*). Quantitative analysis showed that the conformational transition of RecA filaments to the ATP-bound state results in reduction of the average intensity of RecX$_{mNG}$ bound to the tether almost down to a background level (*Figure 4D*).

To address dynamics of the *apo*-ATP transition, we performed continuous fluorescent visualization of the RecA-ssDNA filament during the transfer from the ATP-free to the ATP-containing channel (*Video 1*). To keep a constant stretching force during image acquisition, one of the beads was released from the optical trap, and the tether was stretched by a constant fluid flow adjusted to exert a force of ~3 pN. Upon entry into the ATP-containing channel, the dynamic increase in the tether length was correlated to the gradual loss of the RecX$_{mNG}$ fluorescence signal. These observations suggest that ATP-induced transition between inactive and active states of the RecA filament promotes dissociation of RecX.

We also assesed RecX$_{mNG}$ binding to the active form of RecA-ssDNA using non-hydrolyzable ATP analog, ATPγS. The RecA-ssDNA filament was formed in the presence of 0.5 mM ATPγS, after which was incubated in the channel containing 1 µM RecX$_{mNG}$ and 0.5 mM ATPγS for 30 s, and then was visualized in the channel containing 0.5 mM ATPγS and no proteins. As a result, average intensity of the tether was close to the background level (*Figure 4D*) indicating that RecX$_{mNG}$ did not remain bound to the active RecA-ssDNA filament. Thus we suppose that RecX interaction with active form of RecA-ssDNA filament is much weaker compared to the binding of RecX to *apo* state. Interestingly, in the presence of ATPγS, RecX did not induce any shortening of RecA-ssDNA filaments (*Figure 4—figure supplement 1*) indicating the essential role of ATP hydrolysis in the RecX-induced destabilization of RecA-ssDNA filaments.

## RecX efficiently disassembles RecA filaments on dsDNA

In addition to RecA filaments assembled on ssDNA, we decided to test whether RecX exhibits similar effects on RecA filaments formed on dsDNA. Since, the assembly of RecA-dsDNA filament is impeded at room temperature, the experiment was carried out at 37°C. RecA-dsDNA filaments were assembled by applying a tension of 55 pN to the dsDNA molecule (11,071 bp) in the presence of 1 µM RecA and 1 mM ATP (*Figure 5A*). As a result of RecA binding the length of the dsDNA molecule increased to 5.4±0.1 µm (N=9), which corresponds to ~150% elongation of the B-form of dsDNA. At a force of 3 pN RecA-dsDNA filaments exhibited the length of 4.9±0.2 (N=9), which is slightly lower than the length of RecA filaments formed on equivalent ssDNA at the same force; however, this length was stable over a long period of time.

Preassembled RecA-dsDNA filaments were transferred from the channel containing 1 µM RecA and 1 mM ATP to the channel containing 1 µM RecA, 1 mM ATP, and 200 nM RecX (*Figure 5B*) at a constant tension of 3 pN. Surprisingly, in this case, RecX stimulated a dramatic reduction of the tether length down to the length of bare dsDNA within 30 s. Subsequent reintroduction of the tether back to the RecX-free channel did not produce any elongation of the dsDNA molecule. These observations suggest that RecX is highly efficient at irreversible disassembly of RecA-dsDNA filament. Analysis of the results obtained with 200 nM RecX revealed that the average rate of the filament length reduction is two-order higher for the RecA-dsDNA filaments compared to RecA-ssDNA (*Figure 5C*).

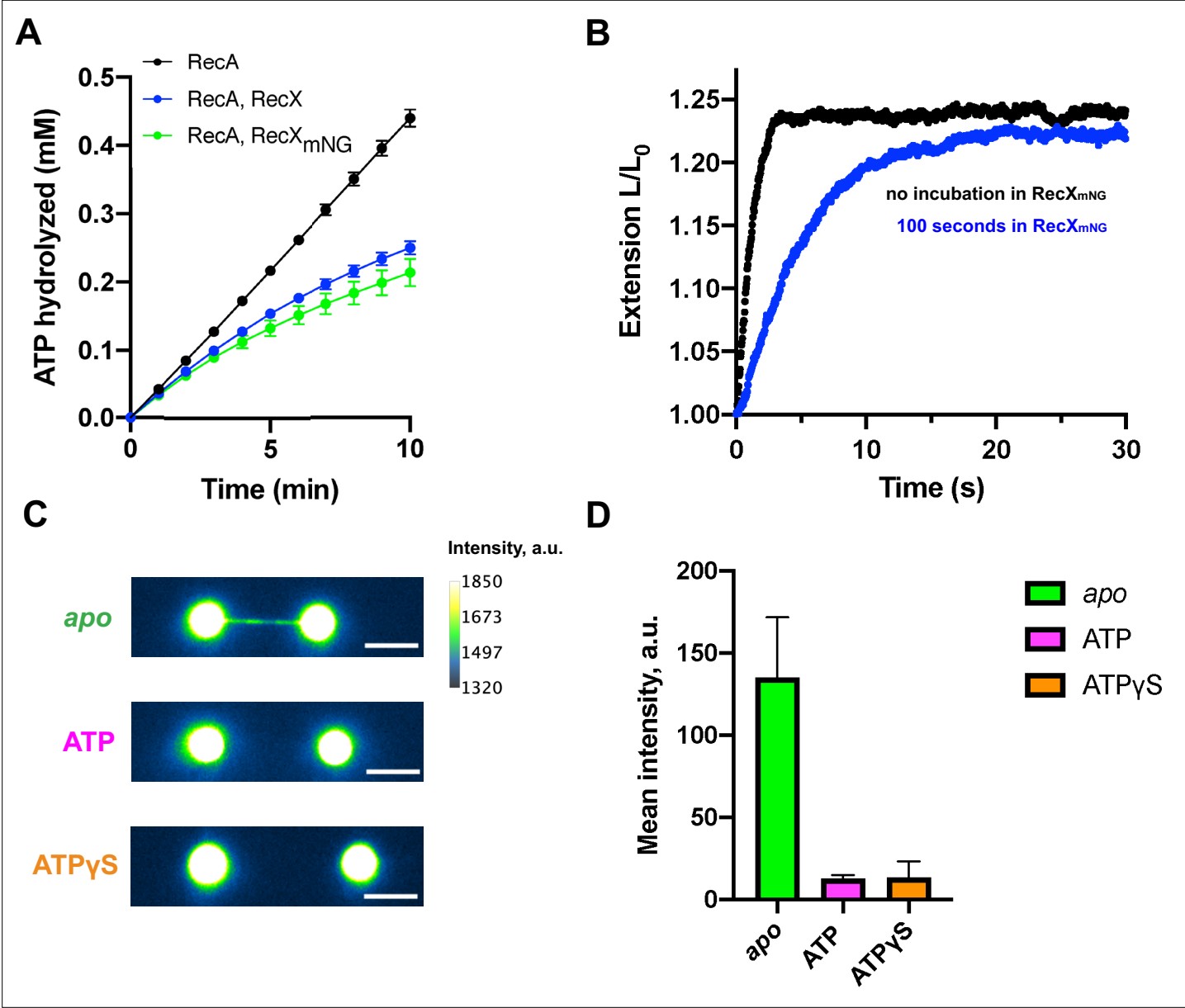

**Figure 4.** Fluorescent visualization reveals that RecX dissociates from the ATP-bound state of the RecA-ssDNA. (**A**) Inhibition of RecA ATPase activity by wild-type RecX (blue) and fluorescent mNeonGreen-RecX (RecX$_{mNG}$) (green). ATP hydrolysis by RecA in the absence of RecX is shown in black. Each data point represents the average of three independent experiments (error bars – SD). (**B**) Relative extension of the RecA-ssDNA filament in the course of *apo*-ATP transition without incubation in RecX$_{mNG}$ (black curve) and after incubation of *apo* RecA-ssDNA filament with 500 nM RecX$_{mNG}$ for 100 s (blue curve). (**C**) Fluorescent images of: RecA-ssDNA filament in *apo* (top) and ATP-bound state (middle) after incubation with 1 μM RecX$_{mNG}$ for 30 s; RecA-ssDNA filament assembled in the presence of ATPgS (bottom) after incubation with 1 μM RecX$_{mNG}$ for 30 s. Scale bar is 5 μm. (**D**) Comparison of the average intensity of the tether after incubation with RecX$_{mNG}$ for *apo* (N=6 molecules), ATP-bound RecA-ssDNA filament (N=3 molecules), and the filament assembled in the presence of ATPgS (N=6 molecules) (consistently with (**B**)). Data are representative of three independent experiments, and values are expressed in mean ± SD.

The online version of this article includes the following source data and figure supplement(s) for figure 4:

**Source data 1.** Source data for RecX$_{mNG}$-induced slowdown decompression of RecA-ssDNA filament and average intensity values for *apo*, ATP-bound, and ATPγS-bound RecA-ssDNA filaments after incubation with RecX$_{mNG}$.

**Source data 2.** Raw tiff images of *apo*, ATP-bound, and ATPγS-bound RecA-ssDNA filaments after incubation with RecX$_{mNG}$.

**Figure supplement 1.** The effect of 1 μM RecX on the dynamics of RecA-ssDNA filament formed in the presence of 0.5 mM ATPγS.

**Figure supplement 1—source data 1.** Source data for *Figure 4—figure supplement 1*.

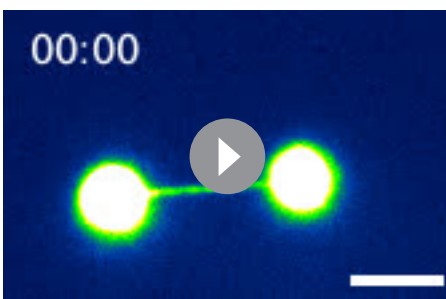

**Video 1.** Continuous fluorescent visualization of the RecX$_{mNG}$-RecA-ssDNA filament during transfer from the ATP-free to the ATP-containing channel. Scale bar is 5 µm.

https://elifesciences.org/articles/78409/figures#video1

## Discussion

The results presented above comprise a detailed analysis of the conformation-specific interactions of the *E. coli* RecX and RecA nucleoprotein filaments. We first discuss the effect of RecX on the RecA-ssDNA filaments. Our experiments confirm that RecX stimulates a net disassembly of the RecA-ssDNA filament; however, this process is very slow and takes hundreds of seconds for an ~5µm long RecA filament to depolymerize. Previously proposed capping model (*Drees et al., 2004a*) suggests that RecX binds to the filament end and blocks its growth leading to a net RecA depolymerization. This model alone fails to explain our results since in the absence of RecA polymerization RecX actively stimulated shortening of the RecA-ssDNA filament (*Figure 2*). Such behavior indicates that RecX also binds along the RecA-ssDNA filament and promotes structural rearrangements and/or disassembly from within the filament.

Unexpectedly, we discovered that RecX interacts with the ADP and *apo* forms of the RecA-ssDNA filament and inhibits their transition into the ATP-bound state. This is in line with the reported inhibition of the RecA ATPase activity by RecX measured in bulk (*Venkatesh et al., 2002*) since depletion of the ATP-bound conformation within the filament should effectively reduce the overall rate of ATP hydrolysis. Interestingly, the possibility of RecX to inhibit RecA ATPase without actual displacement of RecA monomers from ssDNA was proposed previously in *VanLoock et al., 2003a*. It is worth noting that RecX-bound inactive RecA filaments persisted without noticeable changes for over 200 s indicating that RecX does not promote depolymerization of inactive RecA filaments.

Interestingly, when such filaments were supplemented with ATP, RecX gradually dissociated in the course of *apo*-ATP transition (*Figure 4*). Previously, low-resolution electron microscopy demonstrated that RecX can bind along the groove of the ATP-bound conformation of RecA filaments (*VanLoock*

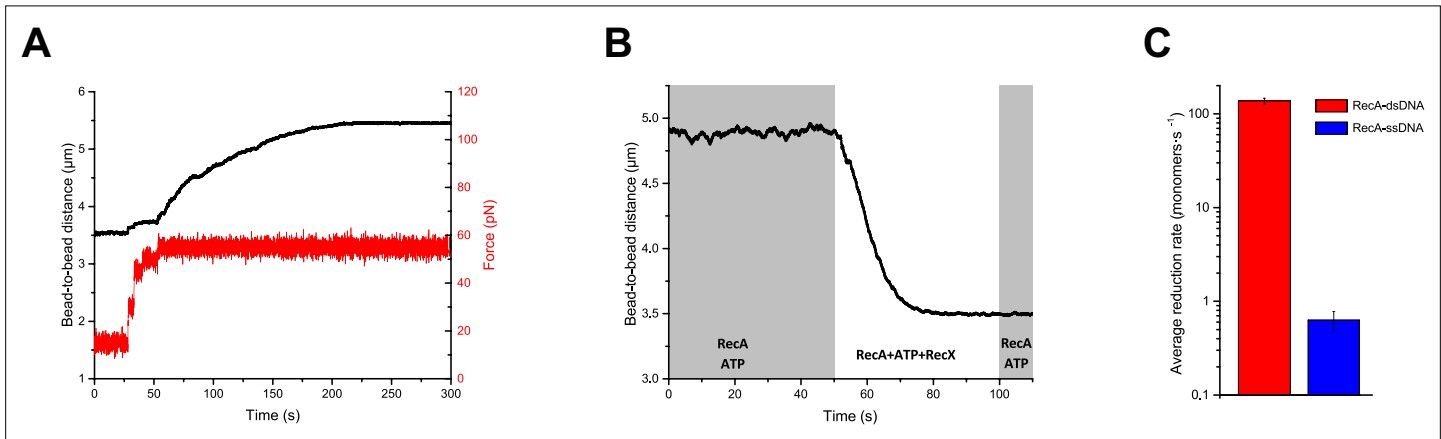

**Figure 5.** RecX effectively promotes disassembly of RecA-dsDNA filaments. (**A**) The assembly of the RecA-dsDNA filament. (**B**) The disassembly of the RecA-dsDNA filament in the presence of 200 nM RecX. (**C**) The comparison of the average length reduction of RecA-dsDNA (N=4) and RecA-ssDNA (N=4) filament induced by 200 nM RecX. The data for RecA-ssDNA is consistent with *Figure 1D*. Data are representative of at least three independent experiments, and values are expressed in mean ± SD.

The online version of this article includes the following source data and figure supplement(s) for figure 5:

**Figure supplement 1.** RecA-dsDNA filament is stable only in the presence of both free RecA and ATP.

**Source data 1.** Source data for RecA-dsDNA filament assembly profile, RecX-induced disassembly of RecA-dsDNA filament, and average length reduction rate for RecA-dsDNA and RecA-ssDNA filaments in the presence of RecX.

**Figure supplement 1—source data 1.** Source data for *Figure 5—figure supplement 1*.

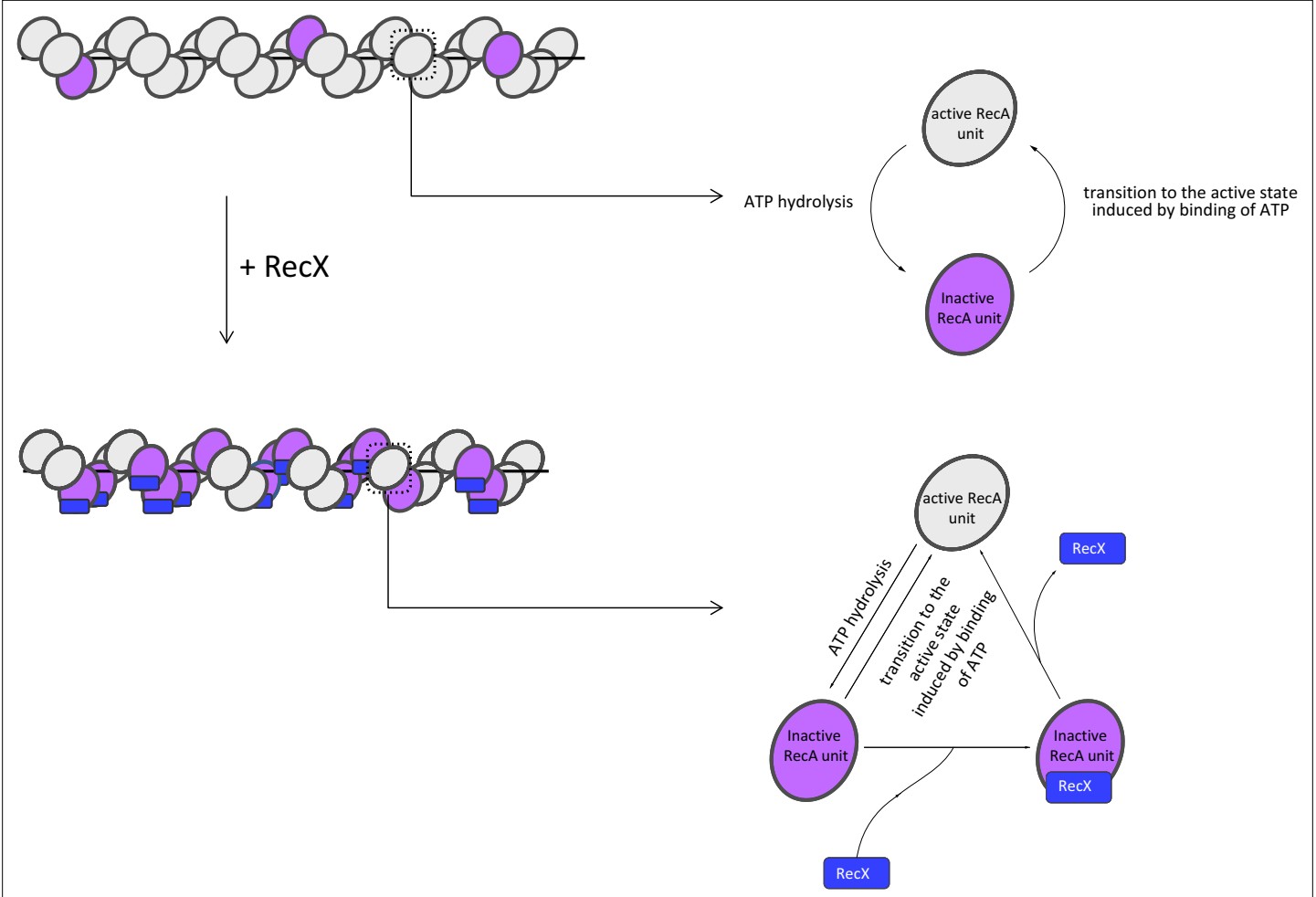

**Figure 6.** Model of RecX interaction with RecA-ssDNA filaments under conditions of continuous ATP hydrolysis. In the presence of ATP, RecX binds inactive patches within RecA-ssDNA filaments and hampers the transition to the active state (see text for details).

*et al., 2003a*); however, these interactions were addressed at a higher concentration of RecX (3 μM). Our results indicate that RecX strongly binds the *apo* conformation of the RecA-ssDNA filament while its affinity to the ATP-bound conformation is significantly lower. Depolymerization of the RecA-ssDNA filament occurred only in the presence of ATP (*Figure 2B*), suggesting that RecX-stimulated RecA dissociation from DNA is coupled to ATP binding or hydrolysis.

Based on the observations discussed above, we propose a following model of how RecX interacts with RecA-ssDNA filaments under conditions of continuous ATP hydrolysis (*Figure 6*).

RecX binds the inactive patches, transiently presented within the active RecA-ssDNA filament, and increases their lifetime by hampering the transition into the active state. The transition of the inactive patch bound by RecX to the active state is accompanied by dissociation of RecX. As a result of the dynamic interaction of RecX with inactive states, their average lifetime and fraction within the filament increase, which leads to the reduction in the filament length. When RecX is removed from solution, bound RecX dissociates as the inactive patches change their conformation to the active, the fraction of inactive states returns to the initial level.

It is noteworthy that previous electron microscopy studies provide a possible explanation of how RecX binding hampers the *apo*-ATP transition of the RecA filament. It was shown that the conformational change of the filament is accompanied by a large movement of RecA's C-terminal domain, which is supposed to be allosterically coupled to the ATPase site (*VanLoock et al., 2003b*). According to low-resolution electron microscopy studies, RecX binds from the C-terminal domain of one RecA subunit to the core domain of another (*VanLoock et al., 2003a*). Thus it was proposed that RecX

inhibits RecA ATPase activity by preventing conformational transition through clamping RecA's C-terminal domain. Although the proposed mechanism is in line with the results of the current study, we believe that additional research is required to elucidate the mechanistic basis of the RecX effect on the conformational transitions of the RecA-ssDNA filament.

The current study showed that RecX-induced fast compression of the RecA-ssDNA tether is characterized by positive cooperativity with the Hill coefficient ≈2 (*Figure 1E*). However, such fast compression is in contrast to the slow binding of RecX to the compressed RecA filament which apparently does not reach saturation even after 100 s incubation with a micromolar concentration of RecX (*Figure 3C*). Previous studies revealed that the *apo*-ATP transition of the RecA-ssDNA flament is highly cooperative (*Kim et al., 2014*; *Alekseev et al., 2020b*) meaning that such transitions within the filament are affected by the conformational state of adjacent RecA monomer-monomer interfaces. We hypothesize that apart from inhibition of conformational transitions by direct binding, RecX may interfere with the cooperativity of the *apo*-ATP transition in the process that involves binding of two RecX monomers.

Unlike RecA-ssDNA complexes, RecX is much more efficient at disrupting RecA filaments formed on duplex DNA. Such filaments mimic postsynaptic RecA-DNA complexes, yet their structure is closely related to the presynaptic RecA-ssDNA filament (*Chen et al., 2008*). Considering structural similarity, we assume that RecX interacts with the RecA-dsDNA filament in a similar manner by trapping transiently occurring inactive states. However, the internal tension of the stretched duplex DNA makes the compressed form of the RecA filament highly unstable. RecX impedes the transition into the stable active conformation, and by this stimulates the dissociation of RecA (*Figure 5B*). Instability of the inactive form of RecA-dsDNA filaments was confirmed by a control experiment which demonstrated that elimination of ATP leads to a complete dissociation of RecA from dsDNA within seconds (*Figure 5—figure supplement 1A*). Alternatively, RecA-dsDNA filaments depolymerization may be due to efficient inhibition of the filament growth by RecX via capping mechanism (*Drees et al., 2004a*). RecA-dsDNA complexes are not stable without free RecA and quickly depolymerize as a result of ATP hydrolysis, indicating that a significant turnover of free and DNA-bound RecA is necessary for maintaining the filament (*Figure 5—figure supplement 1B*).

One potential implication of high efficiency of RecX-induced disruption of RecA-dsDNA filaments is preventing erroneous binding of RecA to a duplex DNA inside the cell. Another interesting possibility is that RecX may interact with the RecA-ssDNA filament and stay bound until the homologous dsDNA is found. Once the strand exchange occurs and the heteroduplex is formed within the filament, RecX stimulates fast dissociation of the postsynaptic complex.

Results of this work provide new mechanistic insights into the RecX-RecA interactions and highlight the importance of conformational transitions of RecA filaments as an additional level of regulation of its biological activity. Regulation of recombination by conformation-specific interactions might be relevant for eukaryotic recombinases Rad51, Dmc1.

# Materials and methods

## Key resources table

| Reagent type (species) or resource | Designation | Source or reference | Identifiers | Additional information |
|---|---|---|---|---|
| Recombinant DNA reagent | prl574-rpoC (plasmid) | This paper | | DNA manipulation, 11 kbp-long substrate; Khodorkovskii Lab, NanoBio, SPbPU, St.Petersburg, Russia |
| Recombinant DNA reagent | pBAD-mng-recx (plasmid) | This paper | | Histag-mNeonGreen-RecX purification (Ampicillin resistance); Khodorkovskii Lab, NanoBio, SPbPU, St.Petersburg, Russia |
| Recombinant DNA reagent | Lambda DNA | New England BioLabs | New England BioLabs:N3011L | |
| Sequence-based reagent | Xbai_L_bio | Alkor Bio | | 5'-CTAGCGAGTGXXXXX-3' (X denotes biotin tag) |
| Sequence-based reagent | SacI_L_bio | Alkor Bio | | 5'-XXXXXCAGTCCAGCT-3' (X denotes biotin tag) |

*Continued on next page*

*Continued*

| Reagent type (species) or resource | Designation | Source or reference | Identifiers | Additional information |
|---|---|---|---|---|
| Sequence-based reagent | SacI_S | Alkor Bio | | 5'-GGACTG-3' |
| Peptide, recombinant protein | Klenow fragment | Thermo Scientific | Thermo Scientific:EP0052 | |
| Chemical compound, drug | pyruvate kinase (from rabbit muscle) | Sigma | Sigma:P1506-5KU | |
| Chemical compound, drug | lactate dehydrogenase (from rabbit muscle) | Sigma | Sigma: L2500-25KU | |
| Chemical compound, drug | Biotin-16dCTP | Jena Bioscience | Jena Bioscience:NU-809-BIO16-S | |
| Chemical compound, drug | ATP | Sigma | Sigma:A7699-1G | |
| Chemical compound, drug | ATPγS | Sigma | Sigma:A1388-25MG | |
| Chemical compound, drug | poly(dT) | Sigma | Sigma:P6905-5UN | |
| Chemical compound, drug | PEP | Sigma | Sigma:P7252-1G | |

## DNA constructs and proteins

A linear 11.1 kbp DNA with biotinylated ends was used for all single-molecule manipulations except fluorescence experiments. This DNA construct was prepared as described previously (*Alekseev et al., 2020b*; *Alekseev et al., 2020a*). Briefly, the biotin labeling was performed by a ligation of the biotinylated oligonucleotides XbaI_L_bio and SacI_L_bio to the plasmid vector prl574 containing insertion of the rpoC gene (*Metelev et al., 2017*) digested with XbaI and SacI restriction enzymes. ssDNA molecules were generated during experiment by a force-induced melting of DNA molecules (*Candelli et al., 2013*).

For fluorescence experiments a longer DNA construct was used. A set of 22 kbp and 24 kbp DNA molecules, enabling generation of ssDNA by force-induced melting, was prepared starting from Lambda DNA (48.5 kbp) as follows. The 3'-ends of Lambda DNA were filled in with biotinylated nucleotides. The reaction contained 1× Tango Buffer (Thermo Scientific), 50 µM dATP, dTTP, dGTP (Thermo Scientific), Biotin-16-dCTP (Jena Bioscience), 7.5 nM Lambda DNA (New England Biolabs), 0.3 units/µl Klenow fragment (Thermo Scientific). After incubation at 37°C for 30 min the reaction was heat inactivated (10 min at 75°C), after which DNA was purified using Bio-Gel P-30 size exclusion spin column (Bio-Rad). Then DNA was digested with SacI (Thermo Scientific) in 1× Tango Buffer and subsequently ligated with the biotinylated oligonucleotide SacI_L_bio at 22°C for 2 hr. A 50:1 ratio of oligonucleotides to DNA overhangs was used. Short complementary oligonucleotide SacI_S was added to increase ligation efficiency (*Horspool et al., 2010*). The reaction was heat inactivated (20 min at 65°C), after which DNA was purified from the excess of oligonucleotides using the Bio-Gel P-30 spin column.

Wild-type EcRecA and EcRecX were purified as described previously (*Drees et al., 2004a*; *Cox et al., 1981*; *Drees et al., 2004b*).

To visualize RecX interaction with RecA-ssDNA filament, a fluorescent RecX_mNG fusion protein was prepared as follows. A genetic construct consisting of 6xHis-tag, short linker (GMASM), mNeonGreen gene, short linker (GGGSGG), and recX (*E. coli*) was cloned into the pBAD/HisB plasmid vector. *E. coli* Rosetta cells were transformed with the resulting pBad_mNeonGreen_RecX vector. Bacteria cells were grown at 37°C. When OD600 reached 0.6, expression was induced by addition of 1 mM isopropyl 1-thio-D-galactopyranoside, after which cells were grown for 3 hr. The cells were harvested by centrifugation and resuspended in a lysis buffer (Na-phosphate buffer, pH 7.5, 500 mM NaCl, 5% glycerol, 10 mM Imidazole, 1 mg/ml Lysozyme). Cells were sonicated for 30 min and were centrifuged for 40 min at 16,000 g. After passing through a 0.4 µm filter the supernatant was loaded onto a HisTrap HP 1 mL column (GE Healthcare). The protein was eluted with a lysis buffer supplemented

with 300 mM Imidazole. The eluted fractions were loaded onto a Superose 6 Increase 10/300 GL column (GE Healthcare) equilibrated with 50 mM Tris–HCl pH 8.0 (4°C), 500 mM NaCl, 1 mM Dithiothreitol. The resulting chromatogram revealed a single peak in the 50 kDa region, corresponding to the monomeric fraction of the mNeonGreen-RecX fusion protein (47.6 kDa). The fractions containing $RecX_{mNG}$ were supplemented with 10% glycerol, frozen in liquid nitrogen, and stored at –80°C. The resulting $RecX_{mNG}$ protein showed the ability to inhibit the ATPase activity of RecA at a similar level compared to the wild-type RecX protein.

## Optical tweezers setup

Custom-built dual optical tweezers were used as described previously (*Alekseev et al., 2020b*; *Pobegalov et al., 2015*). In brief, optical trapping was performed with the Nd:YVO4 1064 nm CW laser (5 W, Spectra-Physics BL-106C), high numerical aperture oil immersion lens (LOMO, 100×, NA=1.25), five-channel microfluidic flow chip (Lumicks), and EMCCD camera (Andor Technology, iXon Ultra 897). The x,y-position of one of the traps was additionally controlled with nanometer accuracy by the mirror mounted on a piezo platform (S-330.80L, Physik Instrumente, Karlsruhe, Germany). The applied force and the end-to-end distance of the DNA tether were measured in real time with 30 ms time resolution by processing images of the trapped beads with a custom-made software designed in LabVIEW. The optical trap stiffness was calibrated by a drag force method using a high-precision piezo stage (P-561.3DD, Physik Instrumente). To apply a constant tension to DNA tether, optical tweezers were operated in a force clamp mode.

Fluorescent images of $RecX_{mNG}$-RecA-ssDNA complexes were obtained with a separate EMCCD camera (Cascade II, Photometrics) using freely available MicroManager software. The fluorescence was excited by the DPSS 473 nm CW laser (100 mW, Lasever, LSR473U) attenuated with a 2.0 OD ND filter. Filter set 10 (Carl Zeiss) was used for separating the fluorescence and the excitation light. Images were collected at 50 ms exposure time and electron multiplier gain of 3000 and saved as TIFF files without compression. Images were further processed in FIJI ImageJ.

## Single-molecule assay

### Buffer solution, experimental conditions, and passivation of the microfluidic chip

All experiments were made in 25 mM Tris-HCl (pH 7.5), 5 mM $MgCl_2$, and 50 mM NaCl. ATP-containing channels were also supplied with 10 U/ml pyruvate kinase and 0.2 mM phosphoenolpyruvate (PEP). In fluorescence experiments, microfluidic channels used for imaging were also supplemented with the oxygen scavenging system: 3 u/ml pyranose oxidase (Sigma-Aldrich), 90 units/ml catalase (Sigma-Aldrich), and 1% glucose. Channels of the microfluidic chip were passivated with 0.5% Pluronic F-127 and 0.1% BSA (*Belan et al., 2021*). Experiments with RecA-ssDNA filaments as well as experiments studying DNA binding activity of RecX were done at 22°C. Experiments with RecA-dsDNA filaments were made at 37°C.

### DNA tether formation and force-induced melting

Two of the five channels of the microfluidic chip (*Figure 1—figure supplement 1A*) were correspondingly fed with a 0.01% solution of a 2.1 μm streptavidin-coated polystyrene beads (Spherotech) and a 15 pM solution of dsDNA (11.1 kbp) with biotinylated ends (*Figure 1—figure supplement 1E*). To obtain a single DNA tether, two beads were initially optically trapped and then moved to the DNA-containing channel, where attachment of the DNA to the beads occured. The attachment of a single DNA molecule to the beads was verified by applying a stretching force of about 20 pN to the tether and controlling its length. Additional check was done during force-induced melting, when a dsDNA molecule exhibited an overstretching plateau at a tension of about 65 pN. To obtain ssDNA, the distance between beads was gradually increased until the tension reached 70–80 pN, after which the tether was incubated under this tension for 10–20 s. Typically, the slight increase in the end-to-end distance and a slight decrease in the tension were observed. After this, the DNA tether was relaxed and its force extension was measured. If the force extension curve followed that of ssDNA, then after incubation of DNA tether under low tension for about 7 s its force extension curve was measured once more to verify that dsDNA was fully melted into ssDNA.

DNA tether formation and ssDNA generation using a longer DNA construct (22 and 24 kbp) were performed analogously except that a lower concentration of DNA fragments was used: 3.5 pM instead of 15 pM.

## Manipulations with RecA-ssDNA filament

### RecA-ssDNA filament formation

The assembly of RecA-ssDNA filaments was performed by applying a stretching force of 12 pN to the ssDNA molecule in the channel containing 1 µM RecA and 1 mM ATP. The applied tension promoted the assembly of RecA filaments by removing the secondary structure of ssDNA. Binding of RecA to ssDNA resulted in the elongation of the DNA tether (*Figure 1—figure supplement 1F*).

### The study of the effect of RecX on the active RecA-ssDNA filament

The effect of RecX on the active RecA-ssDNA filament was examined using the configuration of microfluidic channels presented in *Figure 1—figure supplement 1B*. After the assembly of the RecA-ssDNA filament in the channel supplemented with 1 µM RecA and 1 mM ATP, the applied tension was lowered to 3 pN. Under this tension, the RecA-ssDNA filament was moved to the channel which in addition to 1 µM RecA and 1 mM ATP was also supplemented with RecX of indicated concentration and the change in the end-to-end distance was monitored.

To investigate the reversibility of the change in RecA-ssDNA filament length caused by RecX, another configuration of microfluidic channels was used (*Figure 1—figure supplement 1C*). The RecA-ssDNA filament was assembled in second channel, after which at 3 pN tension the length of the RecA-ssDNA filament was registered during incubation and transitions between third and fourth channels, where the third channel contained 1 mM ATP and the fourth channel contained 1 mM ATP and 500 nM RecX. Importantly, both third and fourth channels contained no free RecA. The transfer of the filament between the midpoints of the adjacent channels took less than 5 s.

## Manipulations with inactive RecA-ssDNA filament

The inactive RecA-ssDNA filament was obtained by transferring the preassembled active RecA-ssDNA filament to the ATP-lacking channel (*Figure 1—figure supplement 1D*). During transitions and incubations a constant tension of 3 pN was applied to the RecA-ssDNA filament. To examine the effect of RecX on the inactive RecA-ssDNA filament, it was transferred to the fourth channel supplemented with RecX of indicated concentration.

To examine the effect of RecX on the conformational transition from the inactive state to the active state, the inactive RecA-ssDNA filament was incubated in the presence of RecX (fourth channel) at 3 pN tension and then was moved to the second channel supplemented with 1 µM RecA and 1 mM ATP, where the increase in the RecA-ssDNA filament length was observed due to conformational transition induced by ATP binding. After the filament length was restored, the RecA-ssDNA filament was again moved to the third channel, which caused a transition to an inactive state. In the third channel the filament was incubated for 30 s, after which the filament was moved to the second channel, and dynamics of restoration of the RecA-ssDNA filament length was registered repeatedly.

## Manipulations with RecA-dsDNA filament

To assemble RecA-dsDNA filament, dsDNA was incubated under 55 pN tension in the presence of 1 µM RecA and 1 mM ATP. After the assembly of the RecA-dsDNA filament, the applied tension was lowered to 3 pN for further manipulations. The measurements of the effect of RecX on the RecA-dsDNA filaments were performed similarly to the experiments with RecA-ssDNA filament.

## RecX$_{mNG}$ fluorescence intensity analysis

To calculate average intensity of the tether after incubation with fluorescent RecX$_{mNG}$ a custom Fiji ImageJ script was used (*Source code 1*). First, mean intensity of the background was calculated by measuring mean gray value of the first eight rows of pixels (8 × 52 pixels rectangle, 1 pixel=142 nm) containing neither the tether nor the beads. This value was further subtracted from all the pixels in the image. To avoid the influence of auto-fluorescence of the beads only a central part of the tether comprising a rectangle of 21 × 6 pixels (for *apo* state) and 31 × 6 pixels (for ATP state) was used to calculate an average intensity of the fluorescent signal.

## Normalization of average length reduction rates of RecA-ssDNA and RecA-dsDNA filaments

Average length reduction rates for RecA-ssDNA and RecA-dsDNA filaments were measured in the units of nm/s (*Figure 5C*) and then were recalculated in terms of RecA monomers per second using normalization factors of 1.30 and 0.49 nm/monomer for RecA-ssDNA and RecA-dsDNA filaments, respectively (see Appendix 1 for details).

## ATPase assay

The RecA's DNA-dependent ATPase activity and its inhibition by wild-type RecX and RecX$_{mNG}$ was measured using a coupled enzyme spectrophotometric assay as previously described (*Yakimov et al., 2017*). The reaction mix contained 25 mM Tris-HCl (pH 7.5), 10 mM MgCl$_2$, 2 mM ATP, 2 mM PEP, 30 U/ml pyruvate kinase, 30 U/ml lactate dehydrogenase, 1.5 mM NADH, 5 µM poly(dT), and 3 µM RecA. The ATPase assay was performed at 37°C. The amount of ATP hydrolyzed was followed by the decrease in the absorbance at 380 nm using the NADH extinction coefficient $\varepsilon_{380}$=1.21 mM$^{-1}$cm$^{-1}$. Reaction mix containing RecA and poly(dT) was incubated for 10 min at 37°C followed by addition of 500 nM of RecX.

## Acknowledgements

We are grateful to Elena Znobishcheva (Peter the Great St Petersburg Polytechnic University, St Petersburg) for her help in the preparation of the DNA construct used for fluorescence experiments. This research was funded by the Russian Science Foundation, grant number [19-74-10049].

## Additional information

### Funding

| Funder | Grant reference number | Author |
| --- | --- | --- |
| Russian Science Foundation | 19-74-10049 | Aleksandr Alekseev<br>Georgii Pobegalov<br>Alexander Yakimov |

The funders had no role in study design, data collection and interpretation, or the decision to submit the work for publication.

### Author contributions

Aleksandr Alekseev, Conceptualization, Data curation, Investigation, Methodology, Software, Visualization, Writing – original draft, Writing – review and editing; Georgii Pobegalov, Conceptualization, Formal analysis, Funding acquisition, Investigation, Methodology, Project administration, Resources, Visualization, Writing – original draft, Writing – review and editing; Natalia Morozova, Alexey Vedyaykin, Investigation, Methodology, Writing – review and editing; Galina Cherevatenko, Data curation, Formal analysis, Methodology, Validation; Alexander Yakimov, Formal analysis, Investigation, Writing – review and editing; Dmitry Baitin, Conceptualization, Investigation, Methodology, Resources, Writing – review and editing; Mikhail Khodorkovskii, Conceptualization, Data curation, Formal analysis, Resources, Supervision, Writing – original draft, Writing – review and editing

### Author ORCIDs

Aleksandr Alekseev ⦿ http://orcid.org/0000-0003-4371-265X
Georgii Pobegalov ⦿ http://orcid.org/0000-0003-0836-0732

### Decision letter and Author response

Decision letter https://doi.org/10.7554/eLife.78409.sa1
Author response https://doi.org/10.7554/eLife.78409.sa2

## Additional files

### Supplementary files
- Transparent reporting form
- Source code 1. Average RecX_{mNG} Intensity Calculation.

### Data availability
Source data files were provided for Figures 1, 2, 3, 4, 5 and Figures supplements. Raw fluorescent images of DNA tether were provided for Figure 4C as Zip file.

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

## Appendix 1

### Normalization of length reduction rates of RecA-dsDNA and RecA-ssDNA

Normalization of length reduction rates of RecA-dsDNA and RecA-ssDNA measured in units of nm/s was performed as follows: the length of 1.44 nm per 1 RecA bound to DNA was calculated from the length of the fully assembled ATP-bound RecA-ssDNA filament under 3 pN tension (5.3 μm) assuming that 11071 nt DNA is bound with a 3:1 stoichiometry. The value of 1.44 nm per monomer was further used for both RecA-ssDNA and RecA-dsDNA filaments. The length of bare ssDNA molecule under 3 pN tension is 0.5 μm yielding 0.14 nm per 3 nt for ssDNA. Thus dissociation of the 1 RecA monomer from the RecA-ssDNA filaments leads to the reduction of the tether length by 1.30 nm. In as much as, the full disassembly of RecA-dsDNA filament results in 3.5 μm end-to-end distance, the length per 3 bp for dsDNA was assessed to be 0.95 nm (under 3 pN tension). Thus, dissociation of 1 RecA monomer from the RecA-dsDNA filament results in the reduction by 0.49 nm. Respectively, factors of 1.30 and 0.49 nm/monomer were used to normalize the length reduction rates of RecA-ssDNA and RecA-dsDNA filaments.

