## [Editor Report]

This paper is of interest to readers in the fields of DNA repair, DNA-protein interactions, and those employing single-molecule techniques. Using single-molecule methods, the authors visualized how RecX, a negative regulator of homologous recombination in bacteria, interferes with the active species in recombination, the RecA nucleoprotein filament. They showed that RecX binds to the RecA filament in its post-ATP hydrolysis state, promotes RecA dissociation from ssDNA, and causes a reversible conformational change of the filament. The latter mode of RecX action is novel and of particular interest. The authors present an interesting model of the RecX-RecA-ATP-ssDNA system.

---

## [Decision Letter]

**Decision letter after peer review:**

Thank you for submitting your article "A new insight into RecA filament regulation by RecX from the analysis of conformation-specific interactions" for consideration by *eLife*. Your article has been reviewed by 3 peer reviewers, and the evaluation has been overseen by a Maria Spies as Reviewing Editor and Kevin Struhl as the Senior Editor. The following individual involved in review of your submission has agreed to reveal their identity: Kailey Cash (Reviewer #1).

Essential revisions:

1) All reviewers and the editor agreed that to confirm the proposed model, the authors should measure the RecX binding to the extended RecA filament stabilized by the ATP-g-S.

2) Please address also the individual comments

*Reviewer #1 (Recommendations for the authors):*

-Line 40: "RecA-dependent SOS-response activation is one of many pathways", currently reads as "one of pathways"

-Line 197: need space between "Figure" and "2"

-Line 304: what buffer channel? The one without ATP or the one with ATP and RecA in which the filament was initially assembled? It's not clear in the text.

-Line 322: mNG should be subscript (RecXmNG) to remain consistent

-Throughout the manuscript apo vs apo is inconsistent.

Lines 321-323, 325, 405, 437, 441, and 639 apo isn't italicized

-Line 401: "noting" It is worth "noting" that RecX-bound…etc

-Lines 556-557: typo with the temperatures, degree symbol should be between the numbers and the C

-Try to stay consistent with decimals vs commas for numbers in graph axes

Example: Figure 1C-E have commas to indicate decimal place while Figure 2B has decimals

*Reviewer #2 (Recommendations for the authors):*

Lines 289 and 298: The authors reference (data not shown) but it is recommended to use a supplement to present such data.

The comparison of length reduction rate for dsDNA and ssDNA in Figure 5C in units of nm/s is combining two separate values of interest that should be displayed separately. First, the length change of dsDNA and ssDNA upon RecA filamentation should be measured (either in nm or preferably fractional extension increase since dsDNA and ssDNA have such different initial lengths). Using this value as a normalization factor, the rate of length reduction can be calculated in units of 1/s, so that underlying kinetic rates of RecX binding and compacting can be directly compared for both cases without the confounding factor of different length scales.

Figure S1 (bare vs RecA) uses two different y axes scales (log and linear) which makes comparing the two curves difficult.

*Reviewer #3 (Recommendations for the authors):*

Overall, this is an interesting study that may be published in *eLife*, provided that the authors satisfactorily address the comments listed below.

1. To test the hypothesis that RecX binds specifically to the inactive conformation of RecA-ssDNA filament (in the apo form), the authors may assemble RecA filament in the presence of non-hydrolyzable ATP analog and test binding of the fluorescently labeled RecX.

2. The authors need to discuss the structural basis of the RecA filament transition between the active and inactive form in the presence of RecX, which does not involve RecA dissociation/reassociation. This transition should strongly affect the filament structure. Is this model consistent with the previous EM studies?

---

## [Author Response]

Reviewer #1 (Recommendations for the authors):-Line 40: "RecA-dependent SOS-response activation is one of many pathways", currently reads as "one of pathways"

Correction in Line 43 was made according to this suggestion of the Reviewer.

-Line 197: need space between "Figure" and "2"

Corrected (Line 217).

-Line 304: what buffer channel? The one without ATP or the one with ATP and RecA in which the filament was initially assembled? It's not clear in the text.

Thank you for this point. The following clarification was made:

Original version:

“..transferred back to the buffer channel to be imaged using wide-field fluorescence microscopy.”

Revised version:

“..transferred back to the buffer channel (lacking both RecX_mNG_ and ATP) to be imaged using wide-field fluorescence microscopy.”

-Line 322: mNG should be subscript (RecXmNG) to remain consistentCorrected (Line 371, Figure 4).-Throughout the manuscript apo vs apo is inconsistent.Lines 321-323, 325, 405, 437, 441, and 639 apo isn't italicizedCorrections were made (Lines 292, 368, 369, 371, 375, 384, 387, 482, 526, 530, 733 of the revised version).-Line 401: "noting" ◊ It is worth "noting" that RecX-bound…etcCorrected (Line 478).-Lines 556-557: typo with the temperatures, degree symbol should be between the numbers and the CCorrected (Lines 648-649).-Try to stay consistent with decimals vs commas for numbers in graph axesExample: Figure 1C-E have commas to indicate decimal place while Figure 2B has decimals

Thank you. In the revised version graph axes are provided with decimals.

Reviewer #2 (Recommendations for the authors):Lines 289 and 298: The authors reference (data not shown) but it is recommended to use a supplement to present such data.

The data regarding RecX interaction with ADP-bound RecA-ssDNA filaments are now provided in Figure 3—figure supplement 3. Results of ATPase measurements are now presented in Figure 4A.

The comparison of length reduction rate for dsDNA and ssDNA in Figure 5C in units of nm/s is combining two separate values of interest that should be displayed separately. First, the length change of dsDNA and ssDNA upon RecA filamentation should be measured (either in nm or preferably fractional extension increase since dsDNA and ssDNA have such different initial lengths). Using this value as a normalization factor, the rate of length reduction can be calculated in units of 1/s, so that underlying kinetic rates of RecX binding and compacting can be directly compared for both cases without the confounding factor of different length scales.

Thank you for this point. We recalculated average length reduction rates of RecA-ssDNA and RecA-dsDNA filaments in terms of RecA monomers per second using normalization factors of 1.30 and 0.49 nm/monomer respectively. Figure 5C was updated and details of the normalization procedure were added to the Materials and methods section:

“Normalization of average length reduction rates of RecA-ssDNA and RecA-dsDNA filaments

Average length reduction rates for RecA-ssDNA and RecA-dsDNA filaments were measured in the units of nm/s (Figure 5C) and then were recalculated in terms of RecA monomers per second using normalization factors of 1.30 and 0.49 nm/monomer for RecA-ssDNA and RecA-dsDNA filaments respectively (see Appendix 1 for details).”

The calculation of normalization factors was provided in Appendix 1.

Figure S1 (bare vs RecA) uses two different y axes scales (log and linear) which makes comparing the two curves difficult.

This figure was updated (Figure 1—figure supplement 2 in the revised version). Force-extension curves are now presented with the same scale.

Reviewer #3 (Recommendations for the authors):Overall, this is an interesting study that may be published in eLife, provided that the authors satisfactorily address the comments listed below.1. To test the hypothesis that RecX binds specifically to the inactive conformation of RecA-ssDNA filament (in the apo form), the authors may assemble RecA filament in the presence of non-hydrolyzable ATP analog and test binding of the fluorescently labeled RecX.

We carried out the experiments with ATP-γ-S, suggested by Reviewers. Figure 4 was updated, and the text covering these results was added in the manuscript:

“We also assesed RecX_mNG_ binding to the active form of RecA-ssDNA using non-hydrolyzable ATP analog, ATPγS. The RecA-ssDNA filament was formed in the presence of 0.5 mM ATPγS, afterwhich was incubated in the channel containing 1 μM RecX_mNG_ and 0.5 mM ATPγS for 30 seconds and then was visualized in the channel containing 0.5 mM ATPγS and no proteins. As a result, average intensity of the tether was close to the background level (Figure 4D) indicating that RecX_mNG_ did not remain bound to the active RecA-ssDNA filament. Thus we suppose that RecX interaction with active form of RecA-ssDNA filament is much weaker compared to the binding of RecX to the apo state. Interestingly, in the presence of ATPγS RecX did not induce any shortening of RecA-ssDNA filaments (Figure 4—figure supplement 1) indicating the essential role of ATP hydrolysis in the RecX induced destabilization of RecA-ssDNA filaments.”

2. The authors need to discuss the structural basis of the RecA filament transition between the active and inactive form in the presence of RecX, which does not involve RecA dissociation/reassociation. This transition should strongly affect the filament structure. Is this model consistent with the previous EM studies?

We added the discussion of the previous EM studies covering the structural basis of RecA-DNA filament conformational transitions in the presence of RecX:

“It is noteworthy that previous electron microscopy studies provide a possible explanation of how RecX binding hampers the apo-ATP transition of the RecA filament. It was shown that the conformational change of the filament is accompanied by a large movement of RecA’s C-terminal domain, which is supposed to be allosterically coupled to the ATPase site [7]. According to low resolution electron microscopy studies, RecX binds from the C-terminal domain of one RecA subunit to the core domain of another [25]. Thus it was proposed that RecX inhibits RecA ATPase activity by preventing conformational transition through clamping RecA’s C-terminal domain. Although the proposed mechanism is in line with the results of the current study, we believe that additional research is required to elucidate the mechanistic basis of the RecX effect on the conformational transitions of the RecA-ssDNA filament.”